# MMHead: Towards Fine-grained Multi-modal 3D Facial Animation

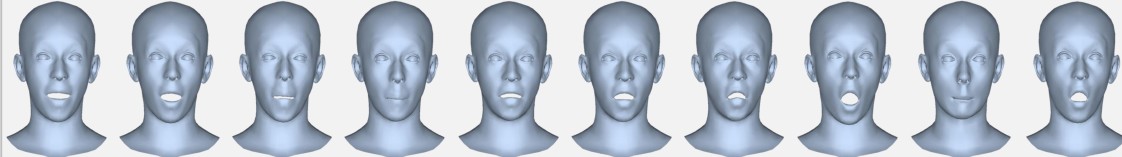

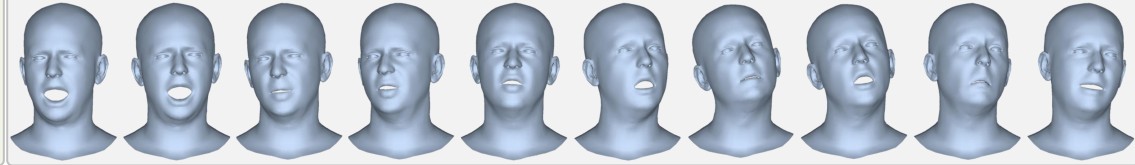

**Figure 1: We present MMHead, the first multi-modal 3D facial animation dataset with hierarchical text annotations including abstract descriptions for overall actions and emotions, and fine-grained descriptions for expressions, head poses, as well as possible scenarios that may cause such emotions.**

## ABSTRACT

3D facial animation has attracted considerable attention due to its extensive applications in the multimedia field. Audio-driven 3D facial animation has been widely explored with promising results. However, multi-modal 3D facial animation, especially text-guided 3D facial animation is rarely explored due to the lack of multi-modal 3D facial animation dataset. To fill this gap, we first construct a large-scale multi-modal 3D facial animation dataset, **MMHead**, which consists of 49 hours of 3D facial motion sequences, speech audios, and rich hierarchical text annotations. Each text annotation contains abstract action and emotion descriptions, fine-grained facial and head movements (*i.e.*, expression and head pose) descriptions, and three possible scenarios that may cause such emotion. Concretely, we integrate five public 2D portrait video datasets, and propose an automatic pipeline to 1) reconstruct 3D facial motion sequences from monocular videos; and 2) obtain hierarchical text annotations with the help of AU detection and ChatGPT. Based on the MMHead dataset, we establish benchmarks for two new tasks: text-induced 3D talking head animation and text-to-3D facial motion generation. Moreover, a simple but efficient VQ-VAE-based method named MM2Face is proposed to unify the multi-modal information and generate diverse and plausible 3D facial motions, which achieves competitive results on both benchmarks. Extensive experiments and comprehensive analysis demonstrate the significant potential of our dataset and benchmarks in promoting the development of multi-modal 3D facial animation.

## CCS CONCEPTS

• **Computing methodologies → Animation**.

## KEYWORDS

3D Facial Animation, Multi-modal Generation, Fine-grained Text Annotation, Dataset and Benchmark, VQ-VAE

## 1 INTRODUCTION

3D facial animation is becoming an increasingly popular topic in computer vision and multimedia due to its numerous applications in the multimedia field such as AR/VR content creation, games, and film production. The strong correlation between speech audio and facial movements makes it possible to automatically generate 3D facial motion (which can then be used to animate avatars) from audio, which will greatly simplify the animation production pipelines. Text, as another commonly used modality in human society, plays an important role in human-controlled AIGC. Recently, text guided image [3, 47], video [49, 62] and human motion [23, 54, 82] generation have achieved pleasing results with numerous multimedia applications. It would also be desirable to generate 3D facial motion

*ACM MM, 2024, Melbourne, Australia*

© 2024 Copyright held by the owner/author(s). Publication rights licensed to ACM.
ACM ISBN 978-x-xxxx-xxxx-x/YY/MM
https://doi.org/10.1145/nnnnnnn.nnnnnnn

| Dataset | Modality | | | | Annotation | | | | | Scale | | Property | | | Acquisition | |
|---|---|---|---|---|---|---|---|---|---|---|---|---|---|---|---|---|
| | | | | | Abstract | | Fine-grained | | | | | | | | | |
| | Motion | Text | Audio | RGB | Act. | Emo. | Exp. | Pose | Scn. | Subj. | Dur. | FPS | Lang. | Repr. | Env. | Tech. |
| BIWI [18] | ✓ | - | ✓ | - | - | L | - | - | - | 14 | 1.44h | 25 | EN | Mesh | | |
| VOCASET [11] | ✓ | - | ✓ | - | - | - | - | - | - | 12 | 0.5h | 60 | EN | Mesh | | |
| MeshTalk [46] | ✓ | - | ✓ | - | - | - | - | - | - | 250 | 13h | 30 | EN | Mesh | | |
| Multiface [68] | ✓ | - | ✓ | ✓ | - | - | ✓ | - | - | 13 | - | 30 | EN | Mesh | Lab | 3D |
| MMFace4D [65] | ✓ | - | ✓ | - | - | L | - | - | - | 431 | 36h | 30 | CN | Mesh | | |
| D3DFACS [10] | ✓ | - | - | ✓ | - | - | L | - | - | 10 | - | 60 | - | Mesh | | |
| CoMA [45] | ✓ | - | - | - | - | - | L | - | - | 12 | - | 60 | - | Mesh | | |
| 4DFAB [6] | ✓ | - | - | ✓ | - | - | L | - | - | 180 | - | 60 | - | Mesh | | |
| 3D-ETF [42] | ✓ | - | ✓ | ✓ | - | L | - | - | - | 100+ | 6.5h | - | EN | Blendshape | Mix | Mono. |
| MEAD-3D [13, 27] | ✓ | - | ✓ | ✓ | - | L | - | - | - | 60 | 38h | 30 | EN | FLAME | Lab | Mono. |
| TEAD [86] | - | ✓ | - | - | - | ✓ | - | - | ✓ | - | - | - | - | Blendshape | - | Gen. |
| TA-MEAD [37] | - | ✓ | ✓ | ✓ | - | ✓ | ✗ | - | ✓ | 60 | 40h | 30 | EN | - | Lab | - |
| **MMHead** | ✓ | ✓ | ✓ | ✓ | ✓ | ✓ | ✓ | ✓ | ✓ | 2K+ | 49h | 25 | Mul. | FLAME | Mix | Mono. |

**Table 1: Comparison of Relevant Datasets. The abbreviations "Act.", "Emo.", "Exp.", "Scn." stand for abstract action and emotion annotations, fine-grained expression (facial movement) annotation, and scenarios that may cause such emotion, respectively. The abbreviations "Subj.", "Dur.", "Lang.", "Pepr.", "Env." and "Tech." refer to subjects, duration, language, representation, and the data acquisition environment and the technology used, respectively. "L" indicates that the dataset has labels rather than text descriptions. "✗" indicates that such annotation exists but is not complete. "EN", "CN", and "Mul." represent English, Chinese, and multiple languages, respectively. "Mono." means the 3D facial motion is reconstructed from monocular video, and "Gen." means the 3D facial motion is generated by neural networks.**

from text descriptions. Further, it would be even more inspiring to achieve fine-grained 3D facial animation under multi-modal signal control. However, due to the lack of relevant datasets, fine-grained multi-modal 3D facial animation has been rarely explored.

In the field of 3D facial animation, impressive progress has been made in audio-driven 3D facial animation [1, 11, 41, 50, 56, 66, 70], making it possible to generate 3D facial motion sequences with synchronized lip movements based on the input talking audio. Moreover, some works [13, 42, 64] have incorporated emotion labels to achieve explicit control over talking emotions. Recently, [37, 86] tried to use text descriptions in natural language rather than emotion labels to guide the facial animation. However, they only consider textual descriptions of emotions during talking, while overlooking the actions other than talking, as well as text descriptions of fine-grained facial and head movements. Such limitations seriously affect the convenience and flexibility of 3D face animation in multimedia applications. A major reason for these limitations is the lack of open-source multi-modal 3D facial animation datasets with fine-grained text annotations, as shown in Tab. 1.

To alleviate the scarcity of multi-modal 3D facial animation datasets, we present MMHead, a large and diverse multi-modal 3D facial animation dataset with rich hierarchical text annotations. To the best of our knowledge, MMHead is the first 3D facial animation dataset with both abstract and fine-grained text annotations, including abstract action and emotion descriptions, fine-grained facial and head movement (i.e., expression and head pose) descriptions, as well as possible scenarios that may cause the person's emotion (see Fig. 1 and Tab. 1). Concretely, we build our 3D facial animation dataset from 2D portrait video datasets [35, 63, 68, 76, 87], considering the huge amount of 2D videos and satisfactory monocular 3D face reconstruction results [12, 19, 20]. An automatic pipeline is also proposed to reconstruct 3D facial motion sequences from monocular videos, and then obtain abstract and fine-grained text

annotations for these facial motions. The data construction pipeline can be easily scaled up to obtain larger datasets. To be specific, we first integrate and filter five public 2D portrait video datasets, i.e., CelebV-HQ [87], CelebV-Text [76], MEAD [63], RAVDESS [35], Multiface [68], to achieve richer facial actions and emotions. Then, we reconstruct 3D facial motion represented by FLAME parameters [32] via a state-of-the-art emotion-preserving monocular 3D reconstruction method named EMOCA [12, 19, 20]. Optimization and manual screening are then performed to obtain final 3D facial motion sequences that can achieve comparable precision compared to the lab-collected 3D facial animation datasets. As for text annotation, we explore the annotation capabilities of the large language model through well-designed prompts. Specifically, the action and emotion labels of the portrait video datasets, per frame activated facial Action Units (AU) [36, 60] and head poses are conditionally combined with five different prompts to feed into ChatGPT [38] to obtain natural text descriptions of abstract action, abstract emotion, detailed expression, detailed head pose, and emotion scenarios, respectively.

Along with the proposed multi-modal 3D facial animation dataset, we benchmark two novel tasks: 1) text-induced 3D talking head animation (Benchmark I), which aims at generating 3D facial animations according to both speech audio and text instructions; 2) text-to-3D facial motion generation (Benchmark II), i.e., generating 3D facial motion sequences based on the given text descriptions. We define these tasks as non-deterministic generation tasks [73, 80] rather than regression tasks [11, 17, 70] to achieve robust and diverse 3D facial motion generation. Moreover, we propose a simple but efficient method named MM2Face to unify the multi-modal information and generate diverse and plausible 3D facial motions, which first utilizes a VQ-VAE [16, 58] to compress the 3D facial motions into a discrete codebook, and then generate plausible results

in a constrained space by sampling the codebook in an autoregressive fashion through a transformer [59] equipped with specially designed attention maps. Based on MM2Face, we further explore the fusion and injection strategies of text and audio modalities to provide insight for future research.

In summary, our main contributions are:

- We propose the first multi-modal 3D facial animation dataset, **MMHead**, which contains rich hierarchical text annotations including abstract action and emotion descriptions, fine-grained facial and head movements descriptions, and possible emotion scenarios.
- We benchmark two new tasks: text-induced 3D talking head animation and text-to-3D facial motion generation to promote future research on multi-modal 3D facial animation.
- We propose a VQ-VAE-based method to unify the multi-modal information and generate diverse and plausible 3D facial motions. The proposed framework achieves competitive results on both benchmarks.

## 2 RELATED WORK

**Audio-driven 3D Facial Animation.** Audio-driven 3D facial animation aims at generating 3D facial motion sequences according to the input speech audio. In recent years, it has attracted much attention due to its potential use in virtual avatar animation [61, 67, 85], film, and game production. Audio-driven 3D facial animation methods can be roughly divided into two categories: rule-based methods [9, 15, 52, 72] and learning-based methods [1, 11, 13, 17, 41, 42, 46, 50, 51, 55, 66, 70]. Rule-based methods [9, 15, 52, 72] typically establish complex mapping rules between pronunciations and lip motions. This kind of method makes it easy to ensure the accuracy of the lip motion, however, requires a lot of manual effort and cannot generate the motion of the entire head. Learning-base methods solve the 3D facial animation problem in a data-driven way, and have made impressive progress in recent years [1, 11, 17, 41, 46, 50, 51, 55, 66, 70]. Furthermore, EmoTalk [42] and EMOTE [13] introduce emotion labels and achieve explicit control over talking emotions. ExpCLIP [86] goes one step further and uses emotion text descriptions rather than labels to control the emotion of audio-driven 3D facial animation. However, it only considers the textual control of emotions during talking, while ignoring the fine-grained control over facial and head movements. In addition, text-to-3D facial motion generation without input speech audio has never been explored except for expression generation [39, 88], which affects the convenience and flexibility of 3D face animation in multimedia applications. To fill this gap and promote the research on fine-grained multi-modal 3D facial animation, we present the first large-scale multi-modal 3D facial animation dataset with rich hierarchical text descriptions in this paper.

**Text-guided Generation.** Incorporating text descriptions into generative models has been a popular topic in AIGC areas such as text-to-image/video/motion generation. In the common setting of text to image task, generative models [21, 28, 31, 58] are widely used to capture the multi-modal distributions of images, and can generate diverse and plausible images compared to the regression models. Early work [25, 30, 53, 57, 69, 78] often uses GANs or VAEs with well-designed networks for text-guided generation. Recently, denoising

diffusion models [5, 14, 47, 49, 54, 74, 77, 79, 82] and VQ-VAE have been explored with superior performance. VQ-VAE is first explored in text to image with numerous researches [4, 16, 75]. In recent years, lots of works [22, 24, 29, 44, 81] also focus on text-to-3D human motion generation and achieve promising results. Despite the impressive progress of text-guided generation in numerous areas, text-guided 3D facial animation has been rarely explored before. To this end, we explore text-to-3D facial motion generation based on the proposed MMHead dataset in this paper.

**3D Facial Animation Datasets.** 3D facial animation datasets play an important role in both researches and multimedia applications, which can be roughly divided into audio-driven 3D facial animation dataset and 3D facial expression dataset. Audio-driven 3D facial animation datasets [11, 18, 46] collects dynamic 3D facial motions with synchronized speech audios and gradually seek to construct larger, higher quality datasets. However, due to the high cost of collecting dynamic 3D data, such datasets are hard to scale up, which has prompted numerous works [13, 27, 42, 51] to reconstruct 3D facial motion from portrait video datasets. Recently, other control signals have begun to emerge in audio-driven 3D facial animation datasets. EmoTalk [42] and EMOTE [13] present datasets with emotion labels. ExpCLIP [86] proposes a text-expression alignment dataset, however, it only contains static facial expressions and ignores facial actions other than talking as well as the corresponding text descriptions. As for 3D facial expression datasets, some works [6, 10, 40, 45, 68, 83, 84] capture someone's 3D face sequences when asking him to make a given expression, which lack of fine-grained text annotations and other facial actions other than the specified expressions. Different from existing 3D facial animation datasets, our MMHead dataset contains both speech audio and rich hierarchical text annotations of diverse facial actions.

## 3 MMHEAD DATASET

Considering the absence of a fine-grained multi-modal 3D facial animation dataset, we present MMHead, a large-scale dataset with both speech audio and fine-grained text annotations. As shown in Fig. 2, we first collect a portrait video candidate set from five publicly available portrait video datasets (Sec. 3.1), and then reconstruct 3D facial motion from this candidate set through monocular 3D face reconstruction technologies [12, 19, 20] followed by optimization and manual check (Sec. 3.2 part 1). In addition, we propose an automatic text annotation pipeline to obtain rich text annotations including abstract action and emotion descriptions, fine-grained facial and head movements descriptions, and three possible scenarios that may cause the person's emotion (Sec. 3.2 part 2). The statistics analysis of the MMHead dataset, and dataset partitioning for the two benchmarks are detailed in Sec. 3.3.

### 3.1 Data Collection

**Dataset Integration.** We integrate five commonly used portrait video datasets with rich facial action and emotion to obtain the candidate video set for our 3D facial motion. (1) **CelebV-HQ** [87], a large-scale and high-quality in-the-wild face video dataset with manually labeled facial action and emotion labels. We remove videos with actions that are hard to see on 3D head, including "kiss", "listen to music", "play instrument", "smoke", and "whisper",

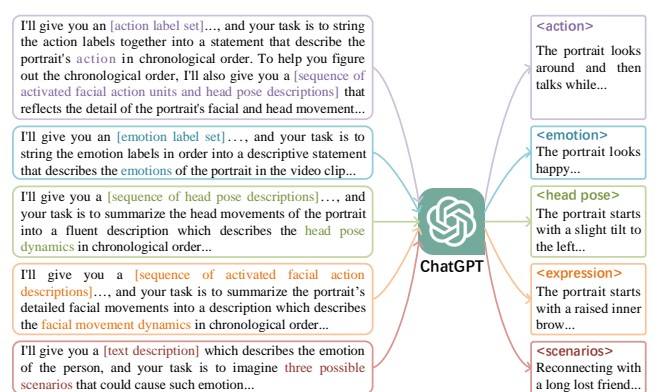

**Figure 2: Dataset construction pipeline. We first integrate five portrait video datasets, and filter the data to obtain the candidate videos for constructing our 3D facial animation dataset. Then, high-precision 3D facial motion sequences are obtained from candidate videos through monocular 3D face reconstruction, FLAME parameter optimization, and manual screening in turn. Finally, we utilize ChatGPT with well-designed prompts to obtain abstract and fine-grained text annotations.**

and reuse the remaining action and emotion labels in our automatic text annotation pipeline for abstract text descriptions. (2) **CelebV-Text** [76], a large-scale and high-quality in-the-wild dataset with facial text-video pairs. Although it has text descriptions for dynamic actions and emotions, it often contains useless descriptions and emotion annotations that change too frequently. To this end, we remove those videos with emotion changes more than four times, and only use the action and emotion labels of the CelebV-Text dataset. Similar to CelebV-HQ, we remove videos with inconspicuous action labels, here "squint" action is also removed. (3) **MEAD** [63], a lab-collected high-quality multi-view emotional talking head video dataset, in which each speaker speaks with 8 emotions in three intensity levels. Here we only use the frontal videos. To balance talking with other actions as well as achieve more obvious emotion labels, we only use the level-3 data for each emotion in the dataset. (4) **RAVDESS** [35], a lab-collected high-quality emotional talking and singing video dataset with 8 emotions. Here we use all of its data. (5) **Multiface** [68], a lab-collected multi-view facial animation dataset with various facial expressions and several talking sequences. Here we only use the frontal videos of recognizable expressions.

**Video Filtering.** We first unify all the portrait videos we got above into 25 FPS, and then remove videos with more than 200 frames. Moreover, we filter out videos with poor audio quality, since the synchronization between audio and facial motion, especially mouth motion, is of great importance, however, out-of-sync sound and picture, excessive background noise or background music, and multiple people speaking together in one video are common in in-the-wild videos. We follow VoxCeleb [7] and utilize SyncNet [8], a pre-trained model for determining the audio-video synchronization, to get the confidence score of each in-the-wild portrait video, and then filter out videos with confidence score lower than a specified threshold.

## 3.2 Automatic Annotation Pipeline

**3D Facial Motion Reconstruction.** We utilize sequences of expression (50 dimensions) and pose (6 dimensions) parameters of a commonly used parametric head model, FLAME [32], to represent 3D facial motion. To reconstruct FLAME parameters from portrait videos, we select a state-of-the-art emotion-preserving monocular 3D face reconstruction method, EMOCA [12, 19, 20], to estimate FLAME parameters for each frame. However, due to the shaking of the people and complex background in the videos, frame-by-frame

**Figure 3: Text annotation pipeline with well-designed prompts. We use different prompts and input information to annotate each data with five types of text descriptions separately by ChatGPT.**

reconstruction results may exhibit instability, such as incorrect shapes and jittery motions. To solve this problem, we first sieve out the outliers of the estimated FLAME parameters through Box Plot and replace them with the values of neighboring points, and then smooth the expression and pose parameters via per-video optimization. All 3D facial motion sequences are manually checked to guarantee quality. For more details, please refer to the Sup. Mat. **Text Annotation.** We annotate each 3D facial motion with rich text descriptions including abstract action and emotion descriptions, fine-grained facial and head movements descriptions, and three possible scenarios that may cause such emotion. These five types of text descriptions are annotated separately as illustrated in Fig. 3. For full content and format of the well-designed prompts, please refer to the Sup. Mat. (1) **Abstract action**. Since the lab-collected part of our candidate video set only contains one action in "talking", "singing" and "making a facial expression", we predefine seven text descriptions for each of these three actions, and then randomly select one of the seven predefined texts as the abstract action text description for each 3D facial motion from lab-collected videos. As for 3D facial motions from in-the-wild videos, multiple actions may occur in one motion sequence and we only have the action labels that are not strictly in chronological order, so we feed ChatGPT [38] additional information including per frame activated AU [36, 60] and head pose labels with their intensity values, and let ChatGPT deduce the chronological order of the action labels, and then give

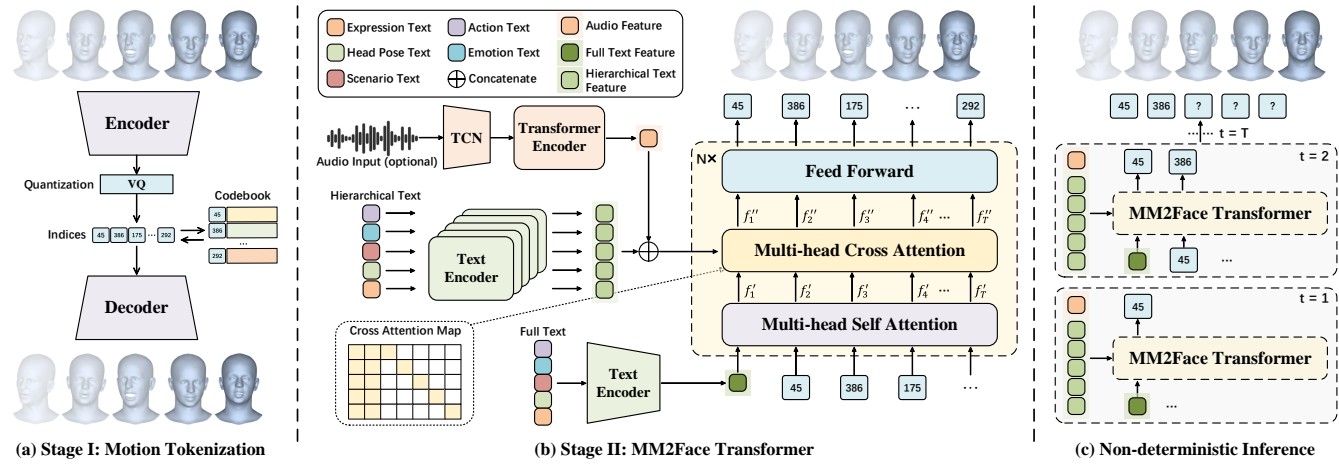

**Figure 4: Overview of our MM2Face framework. In stage I, we utilize a VQ-VAE $\mathcal{V}$ to tokenize the FLAME facial motions $F_{1:T}$ to a sequence of motion tokens. Then in stage II, we utilize a causal auto-regressive transformer MM2Face $\mathcal{G}$ to generate discrete motion tokens $\tilde{F}$ sequentially from audio and text inputs.**

an abstract action text description according to the chronological order. (2) **Abstract emotion**. We let ChatGPT string the given emotion labels (borrow from original video datasets) into a natural text description which will serve as our abstract emotion annotation. (3) **Fine-grained facial movements (*i.e.*, expression)**. We find that letting ChatGPT annotate detailed facial and head movements separately can achieve better performance than annotating them together. Here we feed ChatGPT per frame-activated AU labels together with their intensity values, and ask ChatGPT to summarize them in chronological order to obtain the fine-grained facial movements text descriptions. (4) **Fine-grained head movements (*i.e.*, head pose)**. We define seven types of head pose state labels according to the neck rotation vector of the FLAME [32] model, and feed ChatGPT per frame head pose labels with corresponding rotation angles to obtain the fine-grained head movements text descriptions. (5) **Scenarios**. We finally ask ChatGPT to imagine three possible scenarios that may cause the person to experience the emotions annotated in (2). Note that these hierarchical text descriptions can be automatically annotated in batches, which can be easily extended to larger datasets.

## 3.3 Dataset Analysis

**Data Statistics.** As described above, we construct MMHead from five online portrait video datasets and end up with 35903 3D facial motion sequences up to 49 hours in total. Specifically, there are 10505, 11128, 11313, 2452, 505 motion sequences from CelebV-HQ [87], CelebV-Text [76], MEAD [63], RAVDESS [35], Multiface [68], respectively. Each 3D facial motion is annotated with five types of text annotations including abstract action and emotion, fine-grained facial and head movements, and three possible scenarios. For more details about the motion duration, action and emotion categories statistics, please refer to the Sup. Mat.

**Dataset partitioning for Two Benchmarks.** We benchmark two tasks based on the MMHead dataset. For text-induced talking head animation (Benchmark I), we use a total of 28466 3D facial motion

sequences related to "talk", "sing", and "read". For text-to-3D facial motion generation (Benchmark II), we use 7937 sequences in total, including 505 sequences with various facial expressions from the Multiface dataset, 250 talking, and 250 singing sequences randomly sampled from the data used in benchmark I, and a total of 6932 sequences for all actions other than "talk", "sing", and "read".

## 4 MM2FACE METHOD

We explore a simple but efficient two-stage method **MM2Face** that can handle both newly proposed text-induced 3D talking head animation and text-to-3D facial motion generation tasks. MM2Face can synthesize diverse and plausible 3D facial motions given a speech audio sequence (optional) and a fine-grained text description.

Our MM2Face framework consisting of two stages is illustrated in Fig. 4. In the first stage (section. 4.1), given a 3D facial motion sequence, we utilize a motion tokenizer to extract discrete and compact 3D facial motion tokens. In the second stage (section. 4.2), we conduct an auto-regressive token modeling task in discrete token space using our MM2Face transformer [59]. During inference, MM2Face iteratively predicts the likelihood of motion tokens in each step hence allows for non-deterministic motion generation.

## 4.1 Facial Motion Tokenization

In stage I, we utilize VQ-VAE to extract meaningful discrete facial motion representations. We utilize a 1D CNN-based network [26] to train a facial motion VQ-VAE $\mathcal{V}$ on diverse 3D facial motion data, which consists of a motion encoder $\mathcal{E}$, a vector quantization network $\mathcal{Q}$, a codebook $\mathcal{B}$, a motion decoder $\mathcal{D}$. Given a motion sequence $F_{1:T} = (F_1, F_2, ..., F_T) \in \mathbb{R}^{T \times d}$, where $T$ denotes the motion frame number and $d$ denotes the dimension of FLAME parameters. The encoder $\mathcal{E}$ extract meaningful feature vectors $z = (z_1, z_2, ..., z_{T'})$ from $F$, where $T' = T/r$, $r$ is the temporal downsampling rate ($r = 4$ in our paper). Then the feature vector $z$ is processed by the vector quantization network to get quantized features $\hat{z} = (\hat{z}_1, ..., \hat{z}_{T'})$ via searching nearest neighbors in the

codebook $\mathcal{B}$, which can be written as:

$$\hat{z}_j = argmin_{b_k \in \mathcal{B}} ||z_j - b_k||. \qquad (1)$$

Finally, the quantized feature $\hat{z}$ is fed into the motion decoder $\mathcal{D}$ to reconstruct human motions $F$. Through the process, we can convert facial motion data to discrete features and their corresponding codebook indices $I$, *i.e.*, motion tokens.

**Training Loss.** We adopt the motion reconstruction loss, embedding commitment loss for training. The reconstruction loss is:

$$\mathcal{L}_{rec} = \mathcal{L}_{mse}(F_{1:T}, \hat{F}_{1:T}) + \alpha \mathcal{L}_{mse}(V(F_{1:T}), V(\hat{F}_{1:T})), \qquad (2)$$

where $\mathcal{L}_{mse}$ denotes the MSE loss function, $V(F_{1:T}) = F_{2:T} - F_{1:T-1}$ denotes the velocity of $F$. $\alpha$ is a hyper-parameter.

The commitment loss is:

$$\mathcal{L}_{commit} = ||sg(\hat{z}) - z|| + \beta ||sg(z) - \hat{z}||, \qquad (3)$$

where $\beta$ is a hyper-parameter, $sg$ denotes the stop gradient operation. Finally, we combine these losses $\mathcal{L}_{vq} = \mathcal{L}_{rec} + \mathcal{L}_{commit}$ to optimize the VQ-VAE network.

## 4.2 Auto-regressive Token Modeling

After converting all the facial motions to discrete tokens, auto-regressive token modeling is conducted to train our MM2Face transformer model $\mathcal{G}$. The MM2Face transformer, as shown in Fig. 4, consists of an audio encoder $\mathcal{A}$, a full text encoder $\mathcal{T}_{full}$, hierarchical text encoders $\mathcal{T}_{h1:h5}$ and a MM2Face transformer $C$.

Specifically, the audio $A_{1:T}$ is fed into $\mathcal{A}$ to extract dense audio features, we concatenate all fine-grained text descriptions into a full text and feed it into $\mathcal{T}_{full}$ to obtain full text feature, fine-grained texts are separately fed into hierarchical text encoders $\mathcal{T}_{h1:h5}$ to obtain hierarchical text features, then audio features and hierarchical text features are concatenated together as cross attention inputs. The MM2Face transformer $C$ takes the ground truth motion tokens, audios, full and hierarchical text features as inputs and sequentially predicts the generated motion tokens $\tilde{I}$. Then $\tilde{I}$ can be decoded to final generated facial motions $\tilde{F}$.

**Architecture.** Our audio encoder $\mathcal{A}$ is a pre-trained state-of-the-art speech model wav2vec 2.0 [2], which is composed of a temporal convolution-based network and a multi-layer transformer encoder. $\mathcal{A}$ takes the audio inputs $A_{1:T}$ and output the audio feature. The text encoder $\mathcal{T}$ is a pre-trained distilbert [48] model. The MM2Face transformer $C$ is a multi-layer transformer decoder. Each layer of $C$ is composed of normalization layers, a self-attention layer, a feed-forward layer, and a cross-attention layer.

**Biased Cross-attention.** Giving the concatenated hierarchical text features and audio features as inputs for cross-attention calculation, we specifically design a biased cross-attention mask *mask* in Fig. 4 to align the motion tokens with inputs:

$$Att = Softmax\left( \frac{Q^F(K)^T \times mask}{\sqrt{d_k}} \right), \qquad (4)$$

where $K \in R^{T \times d_k}$ is the key of the concatenated hierarchical text features and audio features.

**Training Loss.** During training, we adopt the teacher-forcing strategy and directly maximize the log-likelihood of the facial motion token representations:

$$L_{MM2Face} = E_{x \sim p(x)}[-logp(x|c)]. \qquad (5)$$

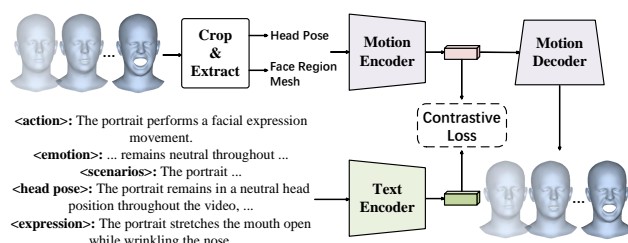

**Figure 5: Overview of text-to-facial motion retrieval method.**

| Dataset | R@1 ↑ | R@2 ↑ | R@3 ↑ |
|---|---|---|---|
| HumanML3D [23] | 0.511 | 0.703 | 0.797 |
| Motion-X [34] | 0.573 | 0.765 | 0.850 |
| Inter-X [71] | 0.429 | 0.626 | 0.736 |
| Inter-human [33] | 0.452 | 0.610 | 0.701 |
| MMHead bench I subset | **0.843** | **0.941** | **0.970** |
| MMHead bench II subset | **0.678** | **0.842** | **0.904** |

**Table 2: Text-to-3D facial motion retrieval performance against other text-to-3D motion datasets.**

**Diverse Probabilistic Facial Motion Inference.** Different from previous deterministic methods [17, 70], our method can generate diverse facial motions with various head poses and expressions given input audio and text descriptions. Specifically, MM2Face iteratively predicts the likelihood of motion tokens in each step, then can stochastically generate the motion token in each step through predicted likelihood.

## 5 EXPERIMENTS

In this section, we first validate quality of our collected MMHead dataset. Then we conduct experiments of MM2Face and other comparison methods on our proposed two benchmark tasks.

### 5.1 Evaluation of MMHead Dataset Quality

**Text-to-3D Facial Motion Alignment Evaluation.** We perform a text-to-motion retrieval task to evaluate the text-to-3D facial motion alignment degree in MMHead dataset. Concretely, we select TMR [43] which is a state-of-the-art retrieval method in 3D human motion area as the baseline method, and calculate the performance on both text-induced 3D talking head animation (benchmark I) and text-to-3d facial motion generation (benchmark II) subsets. We follow the common protocol [23, 33, 34, 71] by using the R-Precision which calculates the average top-1 to top-3 text-to-motion retrieval accuracy in each data batch as evaluation metrics.

Our detailed text-to-3D facial motion retrieval method is shown in Fig. 5. The fine-grained texts from different perspectives are concatenated together and fed into a transformer-based text encoder. The 3D facial motions are cropped and extracted to masked face region mesh sequences and 3D head poses, then are fed into a transformer-based motion encoder.

The results are shown in Table. 2, we compare our text-to-motion retrieval accuracy with other text-to-motion datasets. We can observe that our MMHead dataset subsets achieve the highest accuracy compared to other published text-to-3D human motion

| Methods | Text R-Precision ↑ | | | FID ↓ | Audio-Match ↓ | Diversity → | LVE (mm) ↓ | FVE (mm) ↓ |
|---|---|---|---|---|---|---|---|---|
| | Top-1 | Top-2 | Top-3 | | | | | |
| **Real facial motion** | 0.843 | 0.941 | 0.970 | 0 | 31.70 | 50.84 | 0 | 0 |
| Our VQ-VAE (Recons.) | 0.777 | 0.892 | 0.934 | 14.62 | 35.03 | 50.37 | 2.855 | 0.807 |
| Faceformer [17] | 0.353 | 0.477 | 0.548 | 667.7 | 34.53 | 44.17 | 6.790 | 1.692 |
| CodeTalker [70] | 0.303 | 0.409 | 0.471 | 579.8 | 43.32 | 47.14 | 9.057 | 2.097 |
| Selftalk [41] | 0.286 | 0.399 | 0.478 | 708.4 | 33.19 | 43.23 | 6.766 | **1.680** |
| Imitator [56] | 0.328 | 0.471 | 0.555 | 763.3 | 37.11 | 45.31 | 6.792 | 1.693 |
| Facediffuser [50] | 0.703 | 0.853 | 0.905 | 72.03 | **32.78** | **50.53** | 7.905 | 1.903 |
| Our MM2Face | **0.718** | **0.854** | **0.909** | **41.19** | 34.57 | 50.09 | **6.736** | 1.692 |

**Table 3: Experimental Results for benchmark I: Text-induced 3D Talking Head Animation. The best and runner-up performances are bold and underlined, respectively.**

datasets [23, 33, 34, 71], demonstrating that our text descriptions are accurate enough to conduct multi-modal generation tasks.

**Audio-to-3D Facial Motion Alignment Evaluation.** Similarly, we slightly modify the text-to-motion retrieval method in Fig. 5 by changing the text encoder to a Wav2Vec [2] audio encoder, and removing the 3D head pose as motion inputs. After conducting the experiments, the Top-1 to Top-3 R-Precision accuracy is 0.885, 0.942, and 0.962, demonstrating the great audio-3D face motion matching performance.

## 5.2 Text-induced 3D Talking Head Animation

The provided fine-grained textual annotations combined with corresponding audio 3d facial motions lead to a new task, i.e. text-induced 3D talking head animation. Since there are no methods designed for this task, we select 5 state-of-the-art 3D talking head methods, i.e. Faceformer [17], CodeTalker [70], Facediffuser [50], Imitator [56], Selftalk [41] and modify them by adding a text encoder. Concretely, for mesh-based methods [17, 41, 56, 70], we replace the template feature with the text feature, for parameter-based methods [50], we concatenate the text feature with the audio feature.

**Experiment setup.** We adopt the common protocol to split our dataset into training, validation, and test sets with a ratio of 0.8, 0.05, and 0.15. We select the 56 dimensions FLAME parameters as the input of our method and evaluate the generation performance on generated mesh sequences for fair comparison with other mesh-based methods.

**Metrics.** Previous approaches [17, 70] treat this task as a deterministic prediction task instead of a generative task, hence only the accuracy-related metrics are calculated. To evaluate the diversity and plausibility of generated motions, we follow the previous text to human motion task [23, 24] and select Frechet Inception Distance (FID), text R-Precision, audio matching score, diversity and lip vertices error [17] as our evaluation metrics: (1) **FID**, which is the distribution distance between the extracted features of generated motion and real motion by our pre-trained text-motion retrieval network introduced in Fig. 5. (2) **Text R-Precision**: we calculate the top-1 and top-3 text to motion retrieval accuracy as reported metrics. We specifically compute them by ranking the Euclidean distance between the facial motion and text embeddings in a batch of 32 motion-text pairs. (3) **Audio Matching Score** is the average Euclidean distances between each gt audio feature and the generated motion feature extracted by our audio-motion retrieval network

| Self-atten. | Cross-atten. | Top1-3 R-pre ↑ | FID ↓ | LVE ↓ |
|---|---|---|---|---|
| FT | A | [0.610, 0.776, 0.840] | 45.68 | 6.985 |
| FT, HT | A | [0.714, 0.851, 0.906] | 40.65 | 6.742 |
| FT, HT, A | - | [0.656, 0.798, 0.864] | 72.65 | 8.073 |
| - | FT, HT, A | [0.699, 0.846, 0.908] | 35.57 | 6.824 |
| FT | HT, A | [0.718, 0.854, 0.909] | 41.19 | 6.736 |

**Table 4: Ablation study of different multi-modal information fusion strategies in MM2Face. 'FT', 'HT', 'A' denote full text, hierarchical text and audio respectively.**

trained by the method mentioned in section. 5.1. (4) **Diversity**. We randomly select 300 motion pairs from generated motions. Then We extract motion features and compute the average Euclidean distances of each motion pair to compute motion diversity in test set. (5) **Lip Vertex Error (LVE) [46]** calculates the maximum L2 error across all lip vertices for each frame. (6) **Face Vertex Error (FVE)** calculates the average L2 error across all face region vertices for each frame.

**Quantitative Evaluation.** The quantitative results are summarized in Tab. 3 on MMHead dataset. For motion reconstruction, our VQ-VAE achieves considerable reconstruction performance against the real facial motion, demonstrating the excellent discrete representations. For motion generation, we can observe that our MM2Face achieves the best performance against other state-of-the-art methods in almost all metrics. Concretely, the excellent Text R-Precision accuracy demonstrates that MM2Face is able to synthesize plausible 3D facial motions while correctly following the fine-grained text descriptions. The excellent Audio matching score, LVE and FVE demonstrate MM2Face's ability for precise motion generation. The FID score also indicates the generative distribution modeling performance of our MM2Face method.

**Qualitative Evaluation.** We also provide the qualitative results in Fig. 6. Our approach exhibits superior text and face alignment with accurate audio-visual correspondence. More qualitative results can be seen in supplementary material.

**Ablation Study.** Since our MM2Face method focuses on the multimodal generation of facial motions, we perform an ablation study of various modality fusion strategies. Considering self-attention and cross-attention are the two most popular fusion mechanisms, we conducted a detailed study. The experiment results are summarized in Table. 4. First, we can observe that merely utilizing full text results

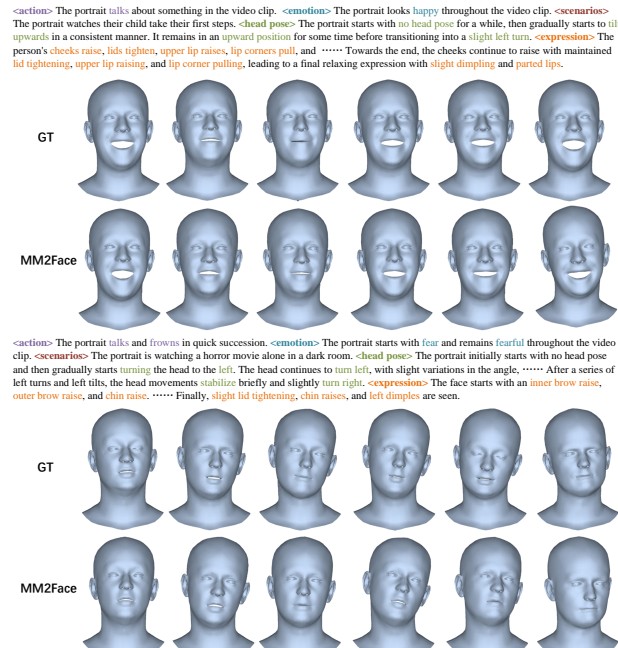

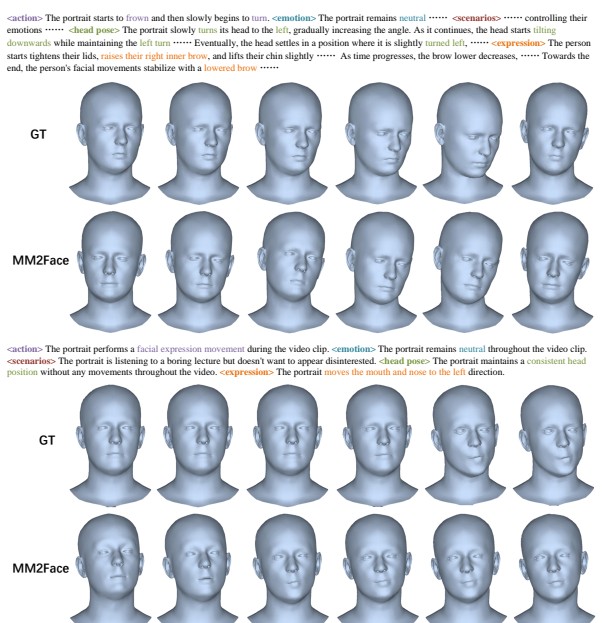

**Figure 6: Qualitative results of MM2Face on benchmark I: text-induced 3D talking head animation.**

**Figure 7: Qualitative results of MM2Face on benchmark II: text-to-3D facial motion generation.**

| Methods | Text R-Precision ↑ | | | FID ↓ | Text-Match ↓ | Diversity → |
|---|---|---|---|---|---|---|
| | Top-1 | Top-2 | Top-3 | | | |
| **Real motion** | 0.678 | 0.842 | 0.904 | 0 | 19.14 | 33.39 |
| VQ-VAE (Recons.) | 0.492 | 0.667 | 0.763 | 21.29 | 21.72 | 33.48 |
| TM2T [24] | 0.316 | 0.481 | 0.559 | 77.21 | 25.35 | 32.17 |
| T2M-GPT [81] | 0.331 | 0.495 | 0.598 | **22.36** | 24.62 | 32.91 |
| MDM [54] | 0.214 | 0.327 | 0.404 | 64.45 | 27.82 | 31.82 |
| Our MM2Face | **0.405** | **0.601** | **0.698** | 35.80 | **23.17** | **33.30** |

**Table 5: Experimental Results for benchmark II: Text-to-3D Facial Motion Generation. The best and runner-up performances are bold and underlined, respectively.**

in poor generation performance, demonstrating the necessity of separating fine-grained texts. Second, fusing all text features and audio features through self-attention mechanism also produces bad results, it is because that self-attention is difficult to model the dense alignment between generated motion tokens and corresponding audio. Third, we can see that the remaining ablations produce slightly similar results but full text self-attention, hierarchical text, and audio cross-attention yield the best R-precision, hence our MM2Face method adopts this architecture.

### 5.3 Text-to-3D Facial Motion Generation

MMHead contains diverse facial expression sequences from specific face expression datasets and general facial motion datasets. Hence we explore a new text-to-3D facial motion generation task. We conduct experiments with the state-of-the-art text-to-3D human motion methods, i.e., MDM [54], TM2T [24] and T2M-GPT [81]. We re-implement these methods to adapt to our dataset format.
**Experiment Setup and Metrics.** Similar to benchmark I, we select 56 dimensions FLAME parameters as motion representations and

adopt the Frechet Inception Distance, R-precision, diversity, and text matching score for evaluation. We re-train a text-to-3D facial motion retrieval network as introduced in Section. 5.1 in benchmark II subset for metric calculation.
**Quantitative Evaluation.** The quantitative results are summarized in Tab. 5 on the benchmark II subset of the MMHead dataset. We can derive that our method achieves the best performance against all metrics. T2M-GPT [81] achieves the second-best performance in FID, matching score, diversity and R-precision. MDM [54] achieves considerable FID performance but fails to capture the text-to-motion consistency according to R-precision. From the results, we derive that our MMHead dataset has the potential for further explorations.
**Qualitative Evaluation.** We also provide the qualitative results in Fig. 7. Our approach exhibits superior text and face alignment. More results are in the supplementary material.

## 6 CONCLUSION

In this paper, we push forward the 3D facial animation task, and present MMHead, the first multi-modal 3D facial animation dataset with rich hierarchical text annotations including abstract action and emotion descriptions, fine-grained facial and head movements descriptions, and possible emotion scenarios. With MMHead, we benchmark two new tasks: text-induced 3D talking head animation and text-to-3D facial motion generation. Moreover, we propose a simple but efficient VQ-VAE-based method named MM2Face to explore the multi-modal information fusion strategies and generate diverse and plausible 3D facial motions, which achieves competitive performance on both benchmarks. We hope that the MMHead dataset and the corresponding benchmarks will promote in-depth research works on multi-modal 3D facial animation.

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
