# OpenReview forum: "MMHead: Towards Fine-grained Multi-modal 3D Facial Animation"
_acmmm.org/ACMMM/2024/Conference — MM2024 Poster_

### Official Review · Reviewer_Tfrz · 2024-05-19

**Rating:** 5
**Confidence:** 2

**Summary:**

The paper introduces MMHead, the first multi-modal 3D facial animation dataset featuring rich hierarchical text annotations. This dataset aims to advance the field of 3D facial animation, particularly in the area of text-guided 3D facial animation, which has been underexplored due to the lack of comprehensive datasets. MMHead provides rich hierarchical text annotations including abstract action and emotion descriptions, fine-grained facial and head movements descriptions, and possible emotion scenarios. With MMHead, the authors benchmark two interesting tasks: text-induced 3D talking head animation and text-to-3D facial motion generation. Furthermore, they propose a simple VQ-VAE-based method to explore the multi-modal information fusion strategies and generate diverse and plausible 3D facial motions.

**Strengths:**

1. The authors present a large-scale, multi-modal 3D facial animation dataset that includes 49 hours of 3D facial motion sequences, speech audios, and detailed text annotations. They provide a comprehensive comparison with other datasets in Table 1.
2. They provide hierarchical text annotations in the dataset for text-guided 3D facial animation. Each annotation in MMHead contains abstract descriptions of actions and emotions, fine-grained descriptions of facial expressions and head poses, and possible scenarios that could elicit such emotions.
3. They propose a VQ-VAE-based method named MM2Face to unify multi-modal information and generate diverse and plausible 3D facial motions, achieving competitive results on the benchmarks.
4. They introduce an automatic pipeline for reconstructing 3D facial motion sequences from monocular videos and obtaining text annotations with the help of AU (Action Unit) detection and ChatGPT, which inspires us a lot.

**Limitations:**

1. Although the MMHead dataset is expansive and multi-modal, the paper does not adequately address its generalizability across varied ethnicities, ages, and facial structures. A more diverse dataset encompassing a broader spectrum of human facial variations could enhance the robustness and applicability of the findings.
2. The VQ-VAE-based method, MM2Face, appears somewhat simplistic and underdeveloped. Qualitative results clearly indicate a significant gap between the synthesized outputs and realistic animation, suggesting that further development and refinement of MM2Face are necessary.

**Suitability:**

3

---

### Official Review · Reviewer_Vmnp · 2024-05-24

**Rating:** 3
**Confidence:** 4

**Summary:**

This paper presents a multi-modal 3D facial animation dataset, namely MMHead. It contains rich hierarchical text annotations including abstract action and emotion descriptions, fine-grained facial and head movements descriptions, and possible emotion scenarios. Then, the paper benchmarks two new tasks: text-induced 3D talking head animation and text-to-3D facial motion generation. Finally, the paper proposes a VQ-VAE-based method named MM2Face to explore the multi-modal information fusion strategies and generate diverse and plausible 3D facial motions.

**Strengths:**

- This paper makes a significant advancement by presenting a multi-modal 3D facial animation dataset with rich hierarchical text annotations, addressing current dataset shortcomings.
- Additionally, the paper proposes a baseline method based on the VQ-VAE model, which generates diverse and plausible 3D facial motions.

**Limitations:**

- The proposed dataset annotation pipeline in the paper is innovative, but there may be two potential issues. Firstly, existing 2D facial datasets, such as MEAD, each video only has one emotional label, meaning the performer only exhibits one emotional state during their expressions. In this case, is there a problem with fine-grained emotional annotation? For example, if only anger is present in the video, fine-grained annotation may identify other emotional states, which would not match the original video tags. Secondly, the annotation pipeline for this dataset relies on monocular 3D face reconstruction, which could be a bottleneck.
- The paper mentions the concept of " Scenarios," but it is not clear from the results what role scenarios plays. So, what exactly is the role of context?
- Datasets like HumanML3D are for text-to-body motion, while MMHead only includes head motion. Therefore, is the comparison in Table 2 fair?
- Table 3 includes results for text-induced audio-driven methods, but there are two issues. Firstly, it is not clear what the performance difference is between results induced by text and those driven solely by audio, whether there is an improvement, and how significant that improvement is. Secondly, the paper modifies the structure of existing audio-driven methods, such as Faceformer. Is such a change justified? Therefore, results for both pure voice-driven and text-induced audio-driven methods should be added.
- There is a small error; at 1:50 in the video, the audio track is missing.

**Suitability:**

3

---

### Official Review · Reviewer_ujHd · 2024-05-26

**Rating:** 4
**Confidence:** 2

**Summary:**

The author introduced MMHead, the inaugural dataset for multi-modal 3D facial animation that features comprehensive hierarchical text annotations. These annotations cover abstract actions, emotions, detailed facial expressions, and potential emotional contexts. Proposed a VQ-VAE based structure to accomplish the two tasks of text-induced 3D talking head animation and Text-to-3D Facial Motion Generation.

**Strengths:**

1) A new dataset construction pipeline is proposed, which realizes 3D annotation and multi-level text annotation from 2D data, and organizes a new dataset, which is helpful for the research of multi-modal 3D facial animation.
2) A new network structure based on VQ-VAE is proposed, which can simultaneously accomplish the two tasks of text-induced 3D talking head animation and Text-to-3D Facial Motion Generation, enriching the multi-modal 3D facial animation research.

**Limitations:**

1) In Tab 3, firstly, I think it is a bit unfair to simply add the text encoder and compare the network with the one proposed in this paper using only text features and audio features to cat, and the lack of visual comparisons with other network results. Secondly, the authors may add some experiment to compare the validity of the rest of the network structure except for the text part, for example, the comparison of the talking head network should have been supplemented with comparisons on commonly used datasets like VOCASET and BIWI.
2) In Fig 4, the audio input is optional, and there is no mention in the text of how the duration of the generated action should be specified when there is no audio input, or anything like that.
3) What is the effect of picking a few of the multiple types of text annotations individually to control the generation of face motions, have you experimented with that?
4) Why choose just 56 dimensions for the FLAME parameter, how was it chosen, does this mean giving up the SHAPE dimension change? Why not use all flame dimensions?\
The contributions of the article are useful, hope the author can answer the above confusion for me.

**Suitability:**

3

---

### Meta-Review · Area_Chair_uvGb · 2024-06-26

**Recommendation:** Accept (Poster)
**Confidence:** 5

**Metareview:**

All reviewers have a consensus decision with BA or WA. The raised questions have been well answered from the reviewer's feedback. The proposed large-scale, multi-modal 3D facial animation dataset is helpful/valuable for research. One reviewer Vmnp still has a question about the fairness of the experiment, and the authors can be updated in the final version.